# Sustainable Diets as Tools to Harmonize the Health of Individuals, Communities and the Planet: A Systematic Review

**DOI:** 10.3390/nu14050928

**Published:** 2022-02-22

**Authors:** Tatianna Oliva Kowalsky, Rubén Morilla Romero de la Osa, Isabel Cerrillo

**Affiliations:** 1Department of Nursing, Faculty of Nursing, Physiotherapy and Podiatry, Universidad de Sevilla, 41015 Seville, Spain; tatiannaok@gmail.com; 2Instituto de Biomedicina de Sevilla, Hospital Universitario Virgen del Rocío, CSIC, Universidad de Sevilla, 41015 Seville, Spain; 3Centro de Investigación Biomédica en Red de Epidemiología y Salud Pública (CIBERESP), Hospital Universitario Virgen del Rocío, 41015 Seville, Spain; 4Alimentta, Think Tank para la Transición Alimentaria, 18320 Santa Fe, Spain; icergar@upo.es; 5Department of Molecular Biology and Biochemistry Engineering, Area of Nutrition and Food Sciences, Pablo de Olavide University, Carretera de Utrera Km 1, 41013 Seville, Spain

**Keywords:** healthy diets, sustainable diets, climate change, global health

## Abstract

Background. Climate change and global health are inextricably linked. Thus, health systems and their professionals must adapt and evolve without losing quality of care. Aim(s). To identify health and environmental co-benefits derived from a sustainable diet and promotion strategies that favor its implementation. Methods. A systematic search for articles published on sustainable diets and human/planetary health published between 2013 and 2020 was conducted on the databases PubMed, Cinahl, Scopus and Trip from 4 to 7 May 2020 in accordance with the PRISMA guideline. Results. A total of 201 articles was retrieved, but only 21 were included. A calorie-balanced diet mainly based on food of plant origin that would allow the attainment of 60% of daily caloric requirements and a low protein intake from animal foods (focusing in fish and poultry) could significantly reduce global morbi-mortality and the dietary environmental impact maintaining a framework of sustainability conditioned by the consumption of fresh, seasonal, locally produced and minimally packaged products. Discussion. The implementation of sustainable diets requires working on the triangulation of concepts of food–health–environment from schools and that is permanently reinforced during all stages of the life by healthcare workers, who should establish the appropriate modifications according to the age, gender and health situation.

## 1. Introduction

It is expected that human-induced global warming will increase the mortality rate by approximately 250,000 deaths between the third and the fifth decade of this century [1]. The United Nations launched in 2015 the Sustainable Development Goals as a universal call to action to end poverty, climate change and inequality by the year 2030. The fact that climate change and global health are inextricably linked means that health systems and their professionals must adapt and evolve to face this challenge without losing quality of care [2].

Since 2008, the attendees of the Consortium of Universities for Global Health (CUGH) discussed the lack of competencies and standardized study plans to guide programs oriented to global health [3], and healthcare professions have progressively become aware of the importance of this challenge at the level of humanity. Currently, there are already initiatives and proposals to include climate change in the curricula of nurses [4]; its inclusion in medical education has also been proposed [5], and the form of food production and consumption and its relationship with the environment has also been fiercely debated by nutritionists [6].

Despite this, there is scarce evidence of the inclusion of the sustainability dimension in the dietary recommendations carried out by the different health professionals, who still focus mostly on clinical outcomes only [7].

Today, living sustainably should be regarded as a healthy habit; therefore, it should be promoted by health professionals through the promotion of public health. However, there is scarce literature on how to specifically promote sustainable eating patterns. Consequently, the content of this review could be a good starting point to inform and encourage further studies.

Healthcare workers, as nutrition counselors, have an essential role in the nutritional education of patients (therapeutic objectives) and communities (preventive objectives), which positions them as a social speaker for the promotion of a healthy and sustainable diet. However, the way of eating not only has an impact on population health but also has an important environmental impact. The food we consume and the ways in which it is produced, packaged or transported from one side of the world to the other have an energy and environmental cost measured in terms of greenhouse emissions (GHEe), use and deterioration of land and water or loss of biodiversity, among others [8].

Sustainable diets have been defined by the Food and Agriculture Organization (FAO) as diets with low environmental impact that contribute to food and nutritional security and a healthy life for present and future generations. This type of diet improves protection and respects for biodiversity and ecosystems, is culturally acceptable, economically fair, accessible, affordable, nutritionally adequate, safe and healthy, and allows the optimization of natural and human resources [8].

The aim of this study is to evaluate the available scientific literature in order to promote sustainable diets in the professional practice of healthcare workers, requiring the following: (1) to determine what environmental impact is derived from each type of food, (2) to identify health and environmental co-benefits derived from a sustainable diet, (3) and to identify promotion strategies that favor the implementation of sustainable dietary patterns.

## 2. Materials and Methods

This is a systematic review with a narrative synthesis approach. In order to guarantee the methodological quality, this review was elaborated following the new version of PRISMA checklist guidelines published in 2020, and its flow chart for new systematic reviews which included searches of databases and registers only [9].

### 2.1. Search Strategy 

A serial search was carried out on the following databases, in the same order in which they are mentioned: PubMed, Cinahl, Scopus and Trip database. We defined two search strategies to obtain the maximum possible number of articles in relation to the objectives set. Both strategies were implemented in each search engine, selecting the one that produced the greatest number of results. Thus, the strategies finally assigned to each search engine were: “Sustainable diet” AND healthy diet for PubMed and Trip database and Sustainab* AND health AND (food OR nutrition OR diet) for Cinahl and Scopus databases. Both strategies were used in the databases, but in each database, only one strategy was optimal (it obtained more articles and contained the articles obtained with the alternative strategy). Therefore, we have reported the optimal strategy for each database.

The strategy descriptors health, food, nutrition and diet are Mesh terms. The key term “sustainable diet” was added using quotation marks to achieve a better targeted search since without them the variety and number of articles was endless. In the corresponding search in Cinahl and Scopus, better results were observed applying the truncation to the term sustainable. In this way, we would include the articles that discussed sustainability in food and had a relation with health.

The search strategy alone yielded an unmanageable number of results for the scope of this study. Therefore, criteria were established to limit the number of studies to be included, ensuring that they were recent and that the subject was consistent with the objectives set. For this purpose, built-in filters were used as help tools for the different databases, and then we resorted to reading the title, abstract or full text through different phases. The filters incorporated in the search engines used allowed directing the search according to the inclusion criteria described below and fundamentally allowed selecting original articles, languages and publication dates: the rest of the criteria were evaluated by the authors. The articles included were published between 2013 and 2020, in English or Spanish, and provided relevant information catering to one or more of our objectives focusing on the health–environment pair. We excluded articles based on animal models or in vitro studies, focused on the transport/distribution of food, management of waste derived from the agri-food system, and those that only dealt with economic aspects or political analysis, and only surveyed the opinion of the population, editorials, letters to the editor, conference reports, reviews, book chapters, conference papers, erratum, books, letters and notes.

The process of inclusion was carried out through a two-round process. In the first round, the process of search and selection of articles was carried out by two of the three authors (TOK and RMR, separately) in two working sessions (4 to 5 May 2020). Both of them applied the exclusion criteria to the initially obtained raw results and identified duplicate articles. The third author was reserved for cases in which there was no consensus between the first two authors, although their participation was not necessary. Once the selection was completed, they were jointly reviewed in a second round and included under joint approval (6 to 7 May 2020).

### 2.2. Extraction of Information 

The articles selected were divided into three parts, each assigned to an author, according to three objectives proposed. Information was extracted regarding the country where the study was carried out, its design and number of subjects included if applicable and the most relevant results that could help answer one or more of the questions formulated in the “Objectives section”. This information was used to compile Table 1. Information was also extracted from journals that published studies including the journal title, year of publication and the best quartile if they were indexed in more than one category. Once completed, each author reviewed the other’s work in order to detect possible errors in data transcription or the absence of relevant information.

The methodology used does not require the participation of human beings and is therefore exempt from evaluation by an ethics committee.

## 3. Results

In total, 201 records were obtained by adding the four databases consulted. The flow chart (Figure 1) shows the registration number per database and its evolution through the selection process. Finally, 21 reports were obtained and each of them was reported in a different study.

Year of publication ranged from 2013 to 2019, although the year with the highest production was 2018 (8/21). All articles selected were included in the first or second quartile of Journal Citations Reports (JCR), and all of them were English-language journals.

Most of the studies were performed with people or secondary data from European countries, with an outstanding participation of Dutch authors, notwithstanding Iran, Japan, Canada, Australia and New Zealand. Three papers involved a multicountry or global approach.

To respond to objective one, seven articles were found, and to respond to objectives two and three, 12 and 13 were selected, respectively, most of them being useful to respond to more than one objective. Table 1 shows the articles selected including the nationality, design and main findings.

Regarding the methods carried out, only six studies were observational studies, mainly cross-sectional and frequently used data generated by surveys and secondary data. The most used methodological design was the creation of predictive models to infer possible changes in the environmental or health impact derived from alternative eating patterns.

## 4. Synthesis and Discussion

Some authors are convinced that health professionals are the key to promoting a healthy and sustainable future through their educational duty. Therefore, incorporating the dimension of sustainability is essential in nutritional counseling; however, a successful educational intervention requires prior training and conceptual mastery of the subject [10,25].

### 4.1. Determining the Weight of Food Types in Environmental Impact

It is essential to be clear on the environmental impact of foods or diets when establishing consumption recommendations. Different environmental-impact indicators have been used that determine in what sense land, water or the atmosphere are affected. Among them, greenhouse gas emissions (GHGe) have been considered a good proxy for this total environmental load, but this is not the only parameter to have in account.

Thus, Black et al. (2015) considered foods that are minimally processed, locally grown/sourced, organic, seasonal, with less/minimal packaging and vegetarian options as environmentally sustainable [11]. Several authors have adopted the mean values of the footprint in the form of environmental from Springmann et al. (2018) so they could be considered as reference values [26]:The GHG footprint is higher for beef and lamb (~30 gCO_2_eq/g), pork (~3 gCO_2_eq/g), eggs, milk, rice and palm oil (~1–2 gCO_2_eq/g).The freshwater footprint is higher for animal-sourced products, sugar, legumes and rice (0.5–1 m^3^/kg).The cropland use is high for legumes, vegetable oils and oil crops, nuts and seeds and animal-sourced products (5–11 m^2^/kg).The nitrogen footprint is high for animal-sourced products, cereals (wheat, rice, maize), oil crops, nuts and seeds and fruits and vegetables (10–50 kg N/kg).

In general, the way to measure GHG emissions derived from consumed food is through data sets prepared by public or private entities using the LCA technique. For example, Temme EH et al. (2015) used an external dataset produced by “Blonk Consultants” [27]. LCA is a technique for assessing the environmental burdens associated with all stages of a product’s life, in this case from farm to fork. As the authors acquired this information from external sources, it is not detailed exactly what this technique consists of.

The studies establish the current diet of the Dutch population based on the Dutch National Food Consumption Survey and in general they all agree that a marked consumption of meat, milk and cheese regardless of age range and sex. Thus, the Dutch diet is rich in SFA and with low consumption of α-linoleic acid, fiber, EPA and DHA. In addition, in women aged 31–50 years, Fe deficiency was observed [20,28].

In the Netherlands, where National recommendations involve food groups included in the Wheel of Five according to Health Council of the Netherlands’s dietary guidelines, GHGe associated with food was studied in women and men aged between 7 and 69 years. Meat and cheese contributed about 40% and drinks (including milk and alcoholic drinks) 20% to GHGe of daily diets. Major differences between high- and low-GHGe diets were in meat, cheese and dairy consumption as well as in soft drinks (girls, boys and women) and alcoholic drinks (men). Of those, differences in meat consumption determined the differences in GHGe most [27]. Subsequently, similar results were published by Van der Kamp in 2018, showing that the reduction of meat during dinner to less than half or the partial or total substitution of unhealthy drinks (soft drinks and alcohol) for water were measures that could significantly decrease the emission of greenhouse gases derived from the Dutch diet [28].

Results have been reported in the same line from Australia where fresh and processed meat were high contributors (33.9%) to the total dietary GHGe; among them, red meat and poultry contributed 18.8% and 10.9%, respectively [19].

Although products of animal origin seem to be those that produce the greatest environmental impact, not all could have the same weight, since some studies show an improvement in the environmental impact when removing meat, but not when removing fish from the diet [20]. From the productive sector, an effort is also being made in innovation to study the intestinal microbiome of ruminants with the aim of raising cattle with lower methane production, although at the moment it is an option under development that has not yet led to definitively consistent results [31].

On the other hand, foods derived from plants (fruits, vegetables, legumes) have been shown to have less impact on the emission of GHGe, which is why they seem to be interesting foods to include in sustainable diets [15,27]. More specifically, Hendrie et al. (2016) observed that fruit (3.5%) and vegetables (6.5%) were the two smallest contributors to total dietary GHGe [19].

The results presented above explain why various authors found a positive correlation of the total dietary GHGe with total energy and total grams of food consumed [19,21].

### 4.2. Sustainable Diet and Health-Environment Co-Benefits

Pollution and environmental deterioration directly affect our health but also the quality of the food we eat [32]. Thus, there is increasing evidence of the high exposure to environmental contaminants to which we are exposed from birth, since many of them accumulate in breast milk [33]. In addition, environmental pollution is related to the emerging appearance of different types of diseases such as those that have an autoimmune basis [34].

If the evolution of climate change continues, it is estimated that by 2050, global food availability will be reduced, triggering health problems such as malnutrition, stunted growth or anemia, which will lead to the death of 529,000 people worldwide. Nevertheless, the model predicts that Asian countries would reach the greatest mortality rate [25].

Springman (2018) has conducted other global predictive models based on three dietary modifications: reduced meat consumption, appropriate caloric intake control and healthy diets defined by the EAT-Lancet Commission on Healthy Diets from Sustainable Food Systems in the guideline recommendations. As a result, the importance of health-environment co-benefits of a sustainable diet became evident (Table 1). Although no strategy proved to be perfect at a global level, they provided partial improvements at different levels depending on the socio-economic development of the countries. Nevertheless, the choice of an appropriate strategy in each case always led to a reduction in premature mortality equal to or greater than 10% [26].

On the other hand, studies have focused on dietary patterns in specific countries. Thus, a study analyzing and modeling the Dutch dietary pattern shows that adapting the current Dutch diet to nutritional recommendations makes it healthier, but not significantly more sustainable. Therefore, it requires making the effort to choose low-GHGe foods to achieve health–environment co-benefits [29].

The health impact of a sustainable diet has also been assessed through the disability-adjusted life years (DALYs) indicator, which reflects the number of healthy life equivalent years lost due to poor health status or disability. In Switzerland, it has been reported that a transition from the current Swiss diet to a pattern better adjusted to national recommendations (guidelines of Swiss society of nutrition) was the most sustainable option involving 36% lesser environmental footprint and 2.67% lower adverse health outcome (DALYs) compared with the current diet. This transition implies a slight increase in fruits and vegetables (from 265 and 239 to 325 and 291 g capita^−1^ day^−1^, respectively) and a more important increase in nuts and legumes (from 5 and 24 to 26 and 50 g capita^−1^ day^−1^, respectively). In addition, it decreased the consumption of roots and tubers (from 230 to 149 g capita^−1^ day^−1^) and a significantly decreased consumption of meat, fish, eggs and vegetable oil (from 129, 15, 25 and 71 to 33, 6, 18 and 26 g capita^−1^ day^−1^). The environmental footprint was quantified, showing a reduction of 54% in GHG emission, 32% in land use and 26%, 33% and 34% in water, nitrogen and phosphorus footprints, respectively [14].

There is also information on health–environment co-benefits in Mediterranean countries. The traditional Mediterranean diet (MD) has been widely listed as a healthy dietary pattern that offers protection against cancer, cardiovascular diseases and cognitive improvements [35]. In Spain, greater MD adherence would reduce GHG emissions (72%), land use (58%), energy consumption (52%), and water consumption (33%) [23]. Even if this adherence were focused only on a university population, environmental improvements could be observed [18]. In France, it would mean a 20% decrease in GHG emission [21] and up to 50% in Italy according to a study conducted with high school students [17].

The traditional Japanese cuisine is characterized by a high consumption of soybean, fish, seaweed, vegetables, fruit, and green tea (always cooking fresh and seasonal producer). However, since 1975, the Western eating pattern has been gaining importance in the Japanese diet, rich in processed and animal-origin food. According to Oita et al. (2018), the optimization of current feeding patterns in Japan toward the traditional Japanese cuisine has resulted in significant environmental benefits in terms of nitrogen footprint (55% reduction) [22]. It has been reported that, in Iran, to decrease the water footprint derived from food production and consumption while maintaining optimal and healthy nutritional status, it was more effective to plan the diet following the Recommended Dietary Allowance (RDAs) from available dietary guidelines [24].

### 4.3. Strategies to Promote Sustainable Diets

#### 4.3.1. Identification of Sustainable Dietary Patterns

High industrialization and the purchase of low-quality raw materials has meant that these types of foods can be produced and marketed at low cost, making them very affordable to low-income consumers [23]. Therefore, it is of interest to assess whether some of the dietary patterns historically associated with different societies or cultures meet the patterns of sustainable diets; if so, it should be promoted.

The substitution of animal foods for those of plant origin as a fundamental step in the transition to a sustainable diet is widely proposed. However, the consumption of meat is culturally rooted, and some authors are committed to educating in the consumption of good quality meat rather than eradicating its consumption [16]. There is sufficient scientific evidence to affirm that vegetarian diets in their variants (vegan, flexitarian, pescatarian, ovolacto-vegetarian) are healthy eating patterns. From an eco-sustainable point of view, it improves many indicators of environmental impact [14,22,26]. Therefore, it should be promoted to spread the adherence of these patterns. However, to adopt a vegetarian and especially vegan eating pattern, it is necessary to have access to a wide variety of plant-based foods, and people must know how to combine them, otherwise they may have certain nutritional deficiencies. On the other hand, psychosocial issues such as the possible stigmatization of this eating pattern can be a barrier to its implementation [36].

The literature frequently refers to the benefits reported by predictive models that totally eliminate meat from consumers’ menus. Thus, we should not lose sight of the fact that these are mathematical models that do not include socio-cultural factors and, therefore, ignore a fundamental indicator for establishing a sustainable dietary pattern: acceptability. In this sense, de Boer et al. (2014) advocate not to remove all meats from the diet, due to its significance in the Dutch diet. Instead, they propose reducing its consumption and, when this is carried out, choosing better-quality meat [16].

The nutritional quality of meat is associated with the saturated fat content. The higher it is, the less healthy it is to eat. However, we normally talk about meat referring to different animal species (pork, beef, lamb, chicken, etc.) without qualifying the importance of the cut (lean parts of a species can be healthier than fatty parts of another one) and the livestock feeding, which can influence its corporal composition [37].

It should be pointed out that there are dietary patterns that are highly acceptable in each country of the world due to their traditional character. Even though they have many differences, they are characterized by a main consumption of plant-based foods, a diet rich in fish consumption, and a minimal, although existing, contribution of meat. Among them, and without prejudice to the existence of other examples, we have been able to identify the Mediterranean diet [12,17,18,23] and the traditional Japanese diet, *washoku* [35]. Both are characterized by a predominance of plant-origin foods (for example, cereals, legumes, olive oil and moderate alcohol, etc.), associated with longevity and listed as UNESCO’s Intangible Cultural Heritage, and both of them have been shown to be sustainable diets in a mathematical modeling study, where the New Zealand diet was also evaluated, which, although it could be optimized to reduce its environmental impact, could have fewer associated health benefits [30]. These types of diets can be easily culturally adapted to geographies where they are not native, thus allowing them to extend their benefits on health and the environment.

Thus, the different patterns of sustainable diets observed seem to have something in common. They allow the establishment of a balanced caloric intake, defined as one in which carbohydrates account for 55–60% of daily caloric requirements, 30–35% of calories from fatty foods and 10–15% of calories from protein foods.

Carbohydrates are provided through foods of plant origin (legumes, farinaceous and fruits) to which vegetables (pigmented colored vegetable, leafy green and cruciferous) are added to increase the consumption of fiber and antioxidants with low calorie intake.

Calories derived from fatty foods come from oily fish, nuts and vegetable oils whose consumption is also associated with important vitamins such as E and D.

Calorie consumption to cover protein needs (0.8 gr/kg/day) would already be partially covered thanks to whole grain cereals, legumes and nuts; therefore, a high consumption of foods of animal origin is not necessary, focusing on fish and poultry due to their low saturated fat content. In fact, it is not even necessary to consume foods of animal origin to cover protein needs, as a correct vegan diet has shown [38]. However, we are aware that, as some authors pointed out, many cultures have meat consumption integrated into their eating pattern [16], so veganism is not the option that best suits these communities.

#### 4.3.2. Promotion in Clinical Practice and at the School Environment

In order to include the dimension of sustainability in the nutritional management of the patient, as well as in the promotion of healthy lifestyles for the general population, it is important that the healthy and sustainable alternatives identified above are listed in clinical practice guidelines and dietary guidelines.

Nutritional counseling is a routine practice to a greater or lesser extent for different health professionals, and particularly for nurses. They should take into account the age, gender and educational level of the people that they are addressing. Each sector of the population may require specific indications since there are different consumption patterns based on the demographic variables [10,13,21].

On the other hand, children have at least one meal a day in their school environment and the behaviors learned at these ages are more strongly maintained over time, turning the school into a strategic point.

Although schools in Australia integrate environmental awareness into their educational programs through gardening activities, 28 of 33 primary and secondary school cafeterias evaluated in Australia sold sugary drinks and unhealthy fatty foods. No schools fully followed nutritional and sustainable quality guidelines [11]. Hence, the school surroundings must be consistent with the educational intervention in order to have a successful promotion of sustainable diets. In this sense, it would be important to pay attention to the content in vending machines and the school menus. Donati et al. (2016) analyzed the menu of Italian schools, observing a drift from the Mediterranean pattern to the Western industrial one [17].

For all of the above, we encourage all healthcare professionals, especially those working in schools, to consider sustainability when carrying out a patient counseling menu review, the promotion of healthy lifestyles and/or applying diet therapy as a therapeutic tool.

It should be noted that the selection of only Spanish and English in the search is associated with an information bias.

Many studies were based on predictive models using publicly available government information. It is difficult to assess the quality of the information derived from this type of record, which is sometimes published in aggregate form. We have not found tools to evaluate the methodological quality and risk of bias of studies based on mathematical models with a predictive character from secondary data.

In addition, this type of study does not take into account cultural factors that affect the acceptance of the proposed dietary pattern. These studies may not be the methodology that provides the most scientific evidence. However, they are plausible when dealing with global health issues due to the magnitude of the facts studied.

The complexity of global health issues makes it difficult to encompass all the variables involved. For example, although meat production has been shown to have a higher environmental cost than vegetable production, eating locally produced meat may be more sustainable than eating vegetables imported from the other side of the globe.

## 5. Conclusions

The Western eating pattern and the consequent food production system not only endangers the health of people but also the planet. It is therefore highly advisable to redirect the habits and lifestyles of consumers, in which healthcare workers involved in nutritional counselling have a great task at hand through educational activities and health promotion.

A balanced calorie diet based on plant-based foods that can meet the needs of most micronutrients, fiber, carbohydrates, and most of fat and protein is ideal. The supply of protein of animal origin (prioritizing poultry and fish and taking care of the quality of the product) should be implemented in small quantities to ensure protein requirements of 0.8 g/kg/day and make the appropriate adaptations according to the level of physical activity. It could significantly reduce global morbidity and mortality associated with chronic diseases, reducing the environmental impact of food production and maintaining a framework of sustainability conditioned by the consumption of fresh, seasonal, locally produced and minimally packaged products.

This often does not mean implementing new eating patterns, but rather returning to those that have been traditionally practiced. For this reason, it is necessary to bet on a sustainable diet, with existing varieties, which allows citizens the possibility of choosing.

Consolidating the implementation of sustainable diets requires working on the triangulation of concepts of food–health–environment from children in schools, and that is permanently reinforced during all stages of the life, both for the healthy and ill, by healthcare workers (HCW), who should establish the appropriate modifications according to the age, gender and health situation.

## Figures and Tables

**Figure 1 nutrients-14-00928-f001:**
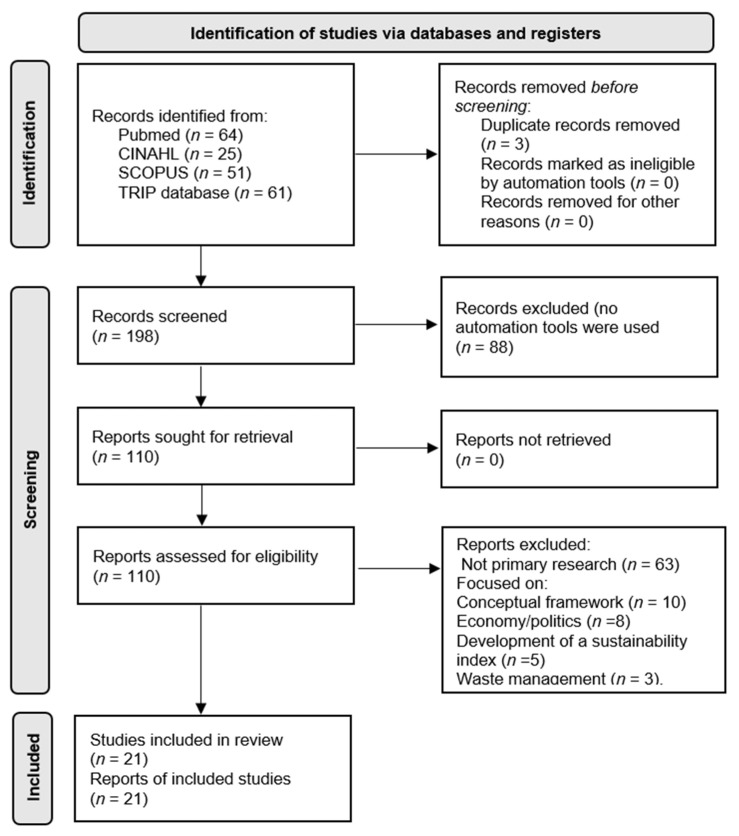
Flow diagram of studies evaluated in the systematic review based on the PRISMA 2020 statement [9].

**Table 1 nutrients-14-00928-t001:** Features and main findings of articles selected.

Reference	Country (N)	Design (Data Source)	Main Findings Related with Nutrition/Health Impact	Main Findings Related with Environmental Impact	Promotion
Benedetti et al., 2018 [10]	Italy	Modelling Multilevel: Temporal Geographical Individual (secondary data: Aspect of Daily Life, survey).	From 1997 to 2012: Daily consumption of pasta, rice and bread decreased from 42.28% to 29.37%. Consumption of red meat fluctuated and then dropped slightly while the consumption of pork meat increased. Eggs, fruit, vegetables and dairy product consumption increased slightly.		Gender (men), age (young) and presence of children in the family had a negative influence in adherence to MD as sustainable eating pattern. Being employed and having a higher educative level were strong predictors of MD adherence. These social factors must be taken into account to address the promotion of sustainable diets.
Black et al., 2015 [11]	Canada (33 schools)	Transversal (directed observation, key informants interviews).		Environmentally sustainable foods are those minimally processed, locally grown/sourced, organic, seasonal, with less/minimal packaging and vegetarian options	They show that policies and programs to promote sustainable diets are not well implemented in schools and identify the general absence of proposals for healthy and sustainable food options in schools, despite the fact that most included sustainability through gardening and compost activities. However, there were still vending machines with processed and sugary foods and beverages.
Blas et al., 2019 [12]	Spain	Modelling (secondary data Governmental datasets about food consumption, production, importations and exportations).	Spain abandons MD for a more caloric, meat and fat diet and with fewer fruits and vegetables, which inevitably affects the health of the population.	This type of diet is less efficient at the water level, so returning to MD would reduce water consumption by about 750 L/person per day and improve water-nutritional efficiency by providing more energy, fiber and nutrients per liter of drinking water.	It is advisable to promote a return to the traditional dietary pattern of the Mediterranean diet.
Brink et al., 2019 [13]	Netherland	Modelling (secondary data offered by Health Council of the Netherlands: national consumption data, National dietary guidelines and dietary reference valued, constraints associated to food groups).	A consumption pattern in line with these guidelines reduces the risk of major chronic diseases and supplies adequate amounts of energy and nutrients. The transition from the current dietary pattern to a sustainable one requires a higher consumption of vegetables, fruits, wholegrain foods, nuts, fish and legumes. A reduction in meat consumption to 500 g per week instead of the current 930 g in men and 615 g in women.	The transition from the current dietary pattern to a sustainable one reduces GHGE by up to 13% for men aged 31–50 years, whereas they increase slightly by 2–5% for women. These results could be improved with a further reduction in meat and replacing it by nuts, legumes and eggs.	Therefore, sustainable diets are also subject to personalized recommendations based on demographic characteristics (age and sex)
Chen et al., 2019 [14]	Global	Modelling (secondary data from FAO food balance sheet. Healthy global diet and food greenhouse gas tax diet were designed in a previous study).	They model the environmental, nutritional, economic and health effects of the transition from the current Swiss diet to nine possible eating patterns (current Swiss diet, Healthy Global Diet, diet of Swiss Society in Nutrition, vegan diet, lacto-ovo vegetarian, lacto-ovo pescatarian, flexitarian, protein-oriented diet, meat-oriented diet and food greenhouse gas tax diet. The transition from the current diet to the diet recommended by the guide of the Swiss Nutrition Society is the most optimal as it would reduce the environmental footprint by 36% and the DALYs by 2.67%	Environmental impact associated with diets was measured (GHGe and freshwater footprints, cropland use and, nitrogen and phosphorous footprints): GHGe was the highest with beef and lamb, followed by pork, eggs, milk, rice and palm oil. Freshwater footprint was high for animal-sourced products, sugar, legumes and rice. Cropland use was high for legumes, vegetable oils, nuts and seeds and animal-sourced products. Nitrogen and phosphorus footprints were highest for animal-sourced products, cereals (wheat, rice, maize), oil crops, nuts and seeds and fruits and vegetables.	
Cobiac et al., 2019 [15]	England, France, Finland, Italy, Sweden	Modelling Scenarios: (1) according nutrition recommendations (NR), (2) diet according NR/without GHGe increased, (3) diets according NR/GHGe reduced in 10%, and (4) diet with GHGe minimised.	Sustainable diets substantially improve the health of the population and consequently life expectancy would increase between 2.3 and 6.8 months per country. Simultaneous reduction of GHGe does not reduce the effect size, and in some cases produces additional health benefits.	Diet reduced in GHGe are those with large reductions in consumption of red and processed meats, salt and fats, and increases in fruits and vegetables and fiber.	
de Boer et al., 2014 [16]	Netherland (1083)	Transversal (non-governmental survey).	On average, the meat consumption was 5.4 days/week with portions from 50 to more than 150 g. A statistical association is shown (with small values of r and R^2^) between meat consumption patterns (frequency, portion size), search for substitutes and sociodemographic variables.		However, taking into account the cultural aspect of the Dutch diet it was concluded that it is better to promote the consumption of good quality meat in small portions and recommended frequency than the total abandonment of it and that the true efficiency of a sustainable diet must be based on its total composition and not just on one type of food.
Donati et al., 2016 [17]	Italy (104)	Modelling (survey, 7 days’ dietary records. European Institute of Oncology database and Barilla Center for Food and Nutrition’s database)).	The observed diet was rich in meat, but very poor in fruits and vegetables and the main sources of energy are bread and substitutes (28%), pasta and rice (20%), sweets (19%) and meat (13%). It was optimized according 3 objectives resulting: Minimum Cost Diet, Environmentally Sustainable Diet (CO_2_e, H_2_O consumption and amount of soil and water to regenerate the resources) and Sustainable Diet (integrates both previous).	The environmental impact was evaluated for a period of 7 days per person taking into account both quantities and frequency of consumption for the different food items. A diet which eliminates meat consumption by substituting legumes involved a reduction of CO_2_e by 50%. The sustainable diet model, may lead to a 51% cut in CO_2_e emissions, 9% reduction in H_2_O consumption and 26% less land needed to regenerate the resources compared to the current diet.	
Fresán et al., 2018 [18]	Spain (20,363)	Cohort (7-d record with a semi-quantitative FFQ (136 food items), nine-item MD index).	Participants with better adherence to MD had higher energy and non-fat/low-fat dairy intake. They consumed more fish and seafood, vegetables, fruits, legumes, cereals and beverages (especially water, red wine and other alcoholic beverage, but less sugar-sweetened sodas). However, the consumption of pastries, eggs and meat (any kind) was lower.	To assess the environmental impact, it took into account the production and processing only and just conventional agriculture processes. The category of “meat and eggs” was the one that caused the greatest environmental impact. MD involved lower land use (−0.71 (95% CI −0.76, −0.66) m2/d), water consumption (−58.88 (95% CI −90.12, −27.64) litres/d), energy consumption (−0.86 (95% CI −1.01, −0.70) MJ/d) and GHG emission (−0.73 (95% CI −0.78, −0.69) kg CO_2_e/d).	The promotion of the Mediterranean diet pattern is an ecofriendly and healthy option that could efficiently help prevent chronic diseases while reducing the environmental impact derived from food production.
Hendrie et al., 2016 [19]	Australia	Modelling (2011–2013 Australian Health Survey data on food consumption, household expenditure data, National GHG Inventory).	They modelled the GHGe from 3 current eating patterns: (1) higher nutritional quality and lower GHGe, (2) lower nutritional quality and higher (3) The average existing Australian adults’ intake (4) eating pattern recommended by Australian Guide to Healthy Eating.	The GHGe associated with food were estimated using environmentally extended input–output analysis. There is a significant positive correlation between total energy and amount of food consumed (total food in grams) with total dietary GHGe. Fruits and vegetables were the two smallest contributors to total dietary GHGe (3.5 and 6.5% respectively), and fresh and processed meat and alternatives (33.9%) were the highest contributors (where red meat contributed 17.6% and chicken 11%)	
Kramer et al., 2017 [20]	Netherland	Modelling (Dutch National Food Consumption Survey 2007–2010).	Analyzing the Dutch diet divided into 4 segments according to sex and age, it was found that all had partial nutrient deficiencies (α-linoleic acid, dietary fiber, EPA and DHA, Fe intakes were too low for one or more groups)	GHGe, fossil energy use and land occupation were used to calculate a weighted score for the overall environmental impact of food products. The model predicts a decreased overall environmental impact when a reduction of meat consumption is applied and in bread, fatty fish and legumes are increased. Eliminating fish and dairy products did not appear to be an effective option, while consumers can substantially reduce the environmental impact of their diet by drinking fewer alcoholic and non-alcoholic beverages.	
Masset et al., 2014 [21]	France (1918)	Transversal (7-d record from French Agency for Food, Environmental and Occupational Health Safety, 2006 Kantar-World Panel purchase database and PANDiet index).	Higher-Quality diets were defined as those with a PANDiet score higher than the sex specific median score. Around 20% of adults had sustainable diets which combined a higher nutritional quality and lower GHGE without increasing the cost.	GHGe values provided by an external consulting firm (Greennext Service) following LCA analysis. Lower-Carbon diets were defined as those with a total diet related GHGE lower than the sex-specific median value. More sustainable diets were those with reduced energy intake and reduced energy density, containing the highest content of plant-based foods, particularly starchy foods. In addition, foods of animal origin and alcoholic beverage consumption was highly associated with dietary GHGe. Introducing these changes to the model meant that approximately one fifth of French adults achieved a sustainable diet with better nutritional quality and GHGe decreased by almost 20% at no additional cost compared to the population average.	
Oita et al., 2018 [22]	Japan	Modelling (secondary data).	From 1961 to 2011, the per capita consumption of protein of animal origin has increased in Japan	In parallel with the protein consumption increased, the nitrogen footprint derived from food has increased a 55%.	It is recommended to return to a traditional Japanese diet for its demonstrated link with the delay of senescence and lower traces of nitrogen than the current one.
Sáez-Almendros et al., 2013 [23]	Spain (6000 households, 840 food service sector centres and 230 institutions)	Transversal (secondary data: FAO food balance sheets for 2007, Household Consumption Surveys of the Spanish Ministry of Agriculture, Food and Environment, MD pattern pyramid).	The Mediterranean diet pattern proposed by the Group of Experts of the Mediterranean Diet Foundation (MDP), the current Spanish consumption pattern (SCP) and western consumption based on the US consumption pattern (WDP) are compared.	For GHG emissions, land use, water and energy consumption, MDP < SCP < WDP was always observed. Greater adherence to MD in Spain would reduce GHG emissions (72%), land use (58%), energy consumption (52%), and water consumption (33%).	
Sobhani et al., 2019 [24]	Iran (695)	Modelling (secondary data: semi-quantitative 168-item food frequency questionnaire, Food-Based Dietary Guidelines for Iran).	From a sustainable diet approach, the water footprint derived from the observed real diet and three hypothetical scenarios are analyzed: (A) usual ingested energy, (B) A + serving recommended in the nutritional pyramid and (C) B + nutrients recommended in RDA. Water consumption was A < C < B < actual diet.	The decrease in water consumption was mainly linked to the decrease in food of animal origin. Scenario A was deficient in micronutrients with respect to the real diet. Nor does B by itself ensure an adequate supply of micronutrients. Scenario C appears to be optimal	
Springman et al., 2016 [25]	Global	Modelling (International Model for Policy Analysis of Agricultural Commodities and Trade, food availability data from the FAO and WHO data on mean BMI).	Diets based mainly on plant foods suppose a healthier eating pattern associated with a greater reduction in available food of 3.2% per person that would suppose more than half a million deaths in the world, mainly related to the decrease in fruit consumption and vegetables.		
Springman et al., 2018 [26]	150 countries	Modelling.	Analysis was carried out focusing on 3 objectives: i) Environmental (25–100% of origin animal foods substituted for vegetables) led to a 12% reduction in premature mortality and a 86% reduction in GHG emissions, mainly in high-income countries; ii) food security (25–100% reduction in underweight, overweight and obesity) reduced premature mortality by around 10% and improved nutrient availability, although with little improvement in environmental impact; iii) public health, evaluating the flexitarian, pescatarian, vegetarian and vegan patterns that all showed to be healthy and reduced mortality between 12 and 22%, also reducing the environmental impact.	Reduction of GHEe was mainly mediated by reducing of meat consumption in diets designed for objectives i) Environmental and ii) Public Health. The lower environmental impact reduction measured by others indicators (cropland, nitrogen and phosphorus) was obtained with diets of environmental objective which involved, moreover, a higher water consumption.	The promotion of sustainable diets must take into account the local socio-economic development, since differences were observed in the results according to the different regions of the planet.
Temme et al., 2015 [27]	The Netherlands(3818)	Transversal (Dutch National Food Consumption Survey 2007–2010, Dutch food composition database, Short Questionnaire to Asses Health enhancing physical activity).	The food consumption was measured on two non-consecutive days, by means of a 24 h dietary recall excluding pregnant and lactating women, institutionalized people and those with language barriers. The average total quantity of foods and drinks consumed was 2.2 (SD 0.6) kg/d, 2.5 (SD 0.8) kg/d, 3.1 (SD 0.9) kg/d and 3.4 (SD 1.0) kg/d for girls, boys, women and men, respectively. Of this, 0.9–1.1 kg/d was from foods and the remaining weight was from drinks.	GHGe values provided by an external consulting firm (Blonk Consultant) following LCA analysis. The habitual GHGE of a day’s consumption in the Netherlands was on average 3.2 kg CO_2_e for girls, 3.6 kg CO_2_e for boys, 3.7 kg CO_2_e for women and 4.8 kg CO_2_e for men. About 40% of the GHGe of daily diets stemmed from meat and cheese, with a similar percentage in girls, boys, women and men. Drinks (including milk and alcoholic drinks) involved 20% to daily GHGe	Sustainable diets are also subject to personalized recommendations based on demographic characteristics (age and sex)
Van de Kamp, et al., 2018a [28]	Netherland (2102)	Transversal (Dutch National Food Consumption Survey, Dutch Food Composition Table (NEVO-Table 2011/3.0), Short Questionnaire to Assess Health enhancing physical activity).	Four scenarios were tested: (1) red/processed meat reduction during dinner by 50–75% (meat was reduced with 85 g/day for men and 59 g/day for women); (2) 50–100% of alcoholic and soft drinks replaced by water; (3) cheese consumed in between meals replaced by plant-based alternatives; (4) two combinations of these scenarios. The different scenarios were compared with diet observed from National surveys which was taken as reference.	GHGe values provided by an external consulting firm (Blonk Consultant). Subjects were stratified by gender and dietary GHG emissions. Scenarios 1 and 2 involved a 15–34% reduction of dietary GHG emissions linked to a reduced saturated fatty acid intake and/or sugar intake, reduced energy and iron intakes and adequate protein intake (for both sex). When snacking on cheese was replaced by plant-based substitutes, as well as the replacement of 50–100% of soft/alcoholic drinks by water, a reduction in GHGe by < 10% was observed.	
Van de Kamp, et al., 2018b [29]	Netherland (2106 only for current diet)	Scenario comparison (secondary data: Dutch National Food Consumption Survey, Dutch Food Composition Table, Wheel of five).	They analyse 3 scenarios: (1) the current Dutch diet (diet observed from National surveys), (2) its version adapted to national recommendations (Wheel of 5) and (3) this same adaptation, but only including foods with relatively low GHG emissions.Consumption of red meat is lower in scenarios 2 and 3, but fish consumption was similar in all scenarios. Consumption of most other food groups in the Wheel of Five is lower in the current diet than scenarios 2 and 3.	Scenario 2 involved a GHGe change by −13% for men aged 31–50 years and +5% for women aged 19–30 years. Scenario 3 involved a GHGe reduction from 28 to 46%. It is shown that the current diet is not the most sustainable and that its mere adaptation to the recommendations may not be enough for it to also be sustainable. It is necessary to make an effort to choose foods with low GHG emissions to achieve the health–environment co-benefits	
Wilson et al., 2013 [30]	New Zealand (16)	Modelling.	Current New Zealand dietary pattern is relatively expensive and unhealthy, with high saturated fat and sodium intakes. Scenarios compared were grouped in: (1) low-cost; (2) low in GHGe and low-cost; (3) ‘‘relatively healthy diets’’ with high vegetable intakes according to Mediterranean/Asian style diets and an Asian, but with cost and GHG constraints; and (4) that included ‘‘more familiar meals’’, potentially more acceptable to New Zealanders. All scenarios showed be healthier than current diet. Scenario 2 had health advantages over the current dietary pattern due to higher vegetable content and less sodium and saturated fat. Optimized diets improved stroke prevention associated with higher presence of polyunsaturated fatty acid vs. saturated fat from meat, lower sodium intake and higher potassium intake. Plant-rich diets also provided benefits against colorectal cancer due to their higher fiber content	The lowest CO_2_ emissions were those derived from Scenario 2 which ranged from 1.31 to 1.9 kg of CO_2_ equivalents per person per day. Scenario 1 was associated with outputs of 2.20 to 4.33 kg of CO_2_ equivalents per person per day. Scenarios 1 and 2 were generally complementary, but a trade-off between increased daily food cost and consuming food associated with lower GHGe was observed due to the reduction in higher GHG foods (i.e, eggs and milk) induce the selection of more expensive alternative foods.	

MD: Mediterranean diet, GHGe: greenhouse gas emissions, DALYs: disability-adjusted life years, r: correlation coefficient, R^2^: determination coefficient. Aims: (1) to determine what environmental impact is derived from each type of food, (2) to identify health and environmental co-benefits derived from a sustainable diet, and (3) to identify promotion strategies that favor the implementation of sustainable dietary patterns.

## Data Availability

Not applicable.

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
