# Peer review of "Sustainable Diets as Tools to Harmonize the Health of Individuals, Communities and the Planet: A Systematic Review"

_nutrients, 2022, doi:10.3390/nu14050928_

Round 1

Reviewer 1 Report

The manuscript is well written with a structure based on three aims. Since these aims build a good story through the different sections, two minor aspects would improve the understanding:

  • Table 1 includes references to the aims which were mentioned in the introduction. Given three described aims, there should no reference to aim number 23 or 123, so it seems that some commas are missing.
  • The discussion would be structured more clearly, if the aims would also include the number as well as the wording from the beginning (e.g. 1. Determining the environmental impact from each type of food ....) 

Author Response

Dear reviewer

You can send responses to your comments as an attachment. The authors are grateful for their contributions and reflections.

Sincerely,

Authors

Reviewer 2 Report

Line 51-53: Please add a citation

Line 70: Please correct: “health care”.

Introduction: impact of diets on the climate change (for example, GHG emissions) has to be briefly included in the introduction.  

Table 1: Please expand the abbreviation DM.

Inclusion of major findings point by point in the main findings column would make the table more interesting.

Instead of major findings, classification based on (if not all) any of the major facets/dimension of sustainable diets such as environmental impact, health/nutrition, food security, and biodiversity would be more valuable.

What do the numbers in the aim column indicate?

Line 150: multicountry  

Line 193-194: A Netherlands GHGe study is mentioned but not the main results or conclusions of the study.

Line 206: what type of alternatives are being discussed in this study? Moreover, discussion (briefly at least) on the environmental impact of meat alternatives is missing in this section. Since impact of meat is well known, studies have focused on meat alternatives, which depending of the constituent materials have varied environmental impact (GHG, water footprint, land use).

Line 209: what are different types of animal derived foods? Which of those have the highest or lowest environmental impact?

Line 233: which strategy is being referred? Do authors refer to the dietary modifications mentioned in the preceding sentence?

Line238-242: without referring to table 1, some details about current Dutch diet and national recommendations could be included. Even in Table 1, what are the national recommendations? wheel of 5?

Line 244-249: what is the composition of the sustainable Swiss diet? Does it include lower consumption of red meats or processed meats and caloric foods as well increased consumption of vegetables? If there is a recommendation then briefly mention the components of the diet.

Line 260-262: Likewise, what are the current dietary patterns and its optimization? Please briefly include.  

Line 269-271: generic statement. While cost is an important dimension, commercialization of food products does not always include low quality raw materials.  

Line 275-280: The determinants and hurdles in the shift towards vegetarian diets could also be discussed.

Line 290-295: Requires a citation.

Line 302: please include prominent plant foods (e.g. cereals, legumes, olive oil, and moderate alcohol etc.) consumed in both diets. While these diets may be sustainable in their settings/regions, the feasibility of adapting to these diets in other regions can make this manuscript more interesting.

Line 350-354: Requires a citation.

This review is well written with very few mistakes. The review also undertakes a robust methodology that uses PRISMA for the review of literature. However, the major drawbacks of this manuscripts are as follows:

While it is stated as a main objective to determine the environmental impact of each food, environmental impact of food types section mainly suggests that meat consumption has higher environmental impact than plant based foods. This is already well known. Moreover, deeply entrenched meat eating habits or meat attachment, which are major challenges for meat replacement or reduction are not discussed. Studies have shown that encouraging consumers to less meat can also produce undesirable results. The challenges associated with adaptation of plant based diets (vegetarian, pescatarian, flexitarian, vegan, plant protein oriented) are also not discussed. The only table in the manuscript lists findings from several publications making it difficult to parse information. The strategy section identifies two sustainable diets but fails to include strategies or challenges in the adoption of similar diets in other regions. More importantly, this manuscript recapitulates findings, which have been extensively discussed in numerous publications and fails to provide new insights.

Author Response

Dear reviewer

Reviewer 3 Report

Thank you for granting me the opportunity to review this piece of work. In this work, Kovalsky et al. conducted a systematic review that sought to highlight the potential impact of sustainable diet consumption on human and planetary health as well as the strategies that could be targeted to promote its consumption.  Kindly, find below my comments for your response.

Abstract:

Line 21-22: The authors should kindly indicate the “date” the article extraction was carried out. The total number of articles retrieved during the article searching process and the final articles selected for this systematic review must be indicated. The authors should kindly revise the writing under the “Method section”. They could consider putting it this way “A systematic search for articles published on “sustainable diets and human/planetary health” was conducted on the databases “indicate the list of databases here” on “indicate the date of the search” in accordance with the PRISMA guideline”.

Line 22: The authors should kindly explain this statement “A calorie-balanced diet based on vegetable…” It appears vague and unclear in its present state. Is the authors use of “calorie-balanced diet” in reference to “carbohydrate-based” staples from root and tubers or cereals and grains which should have some adequate amount of vegetables such as that indicated in the “MyPlate” guideline used in the United States of America?

Line 23: What do the authors mean by “slight” protein? It’s important that they indicate the quantity of animal protein. This could add a bit of strength to this review. The average protein requirement for a healthy adult is around 0.8g/kg bodyweight. Does the slight means they should reduce this? If yes, what are the sustainable ways they could meet their protein requirement? Also, what were the predominant “class” of “vegetables” that consumers would need to prioritise? There are different classes of vegetables and so stating the type of vegetables consumers should focus on enhances the quality of the work.

Line 24-25: The authors should indicate the magnitude of reduction for morbi-mortality and the dietary environmental impact. This could strengthen the work.

Line 30: Kindly, add a “Concluding remark” to the Abstract. The essence of a systematic review is to predominantly identify gaps in knowledge or share some new insights on already published works on the subject matter under consideration which the authors could thus recommend to the reader.

Introduction

Line 70: Kindly correct “car” to “care”.

Materials and Method

Line 81: The authors should please indicate the search date for repeatability sake. How many authors did the article search? In instances where there was misunderstanding on whether an article should be added or rejected, which of the authors came into resolve that? Indicating these strengthen the quality of the review.

Line 87-89: I am wondering why the authors did not apply the same “search terms” for all the databases used. The title of this manuscript contains planetary health. However, I find it difficult to understand why the authors had no “planet and planet-related” word in their search terms. Yet, the impact of sustainable diet on the planet and even climate change is one of the outcomes of interest in this review. This could be a limitation of this review. Could they provide a reason for that? They should kindly introduce a “full stop” at the end of the “databases”.

Line 98: The use of “comma” should rather be “full stop”.

Line 101: The authors should please indicate the specific filters that were employed to reduce the number of papers that were obtained after the initial search. Did the filter selectively excluded “review papers”, “reports” etc?

Line 106-111: The authors have indicated the exclusion criteria employed. Were the articles excluded during the “article screening stage” by the authors or the exclusion was carried out by the “filters?” This is important for the reader to appreciate what exact filters were employed.

Line 112-119: The authors should please indicate the “initials” of the names of the authors who were involved in the processes stated there. For example in Line 114, which of the 2 authors were the authors referring to? If the authors had used a reference manager such as ENDNOTE™ for example, sorting of duplicate papers would have been very easy. Does this mean the authors did the whole article extraction (selection and obtaining the articles) manually?

Line 125: The authors should put Objectives section in inverted commas ie. “Objectives section”.

Results

Line 145: I suggest that the authors revise their PRISMA guideline flow chart (Figure 1). I suggest they follow this paper by “Liberati et al. (2009). The PRISMA statement for reporting systematic reviews and meta-analyses of studies that evaluate health care interventions: explanation and elaboration. Journal of clinical epidemiology, 62(10), e1-e34.” This captures the standard the flow diagram for systematic review reporting. They should sum all the papers selected into one “box”. Take out the duplicate papers. The rest of the guidelines are captured in the reference I have suggested above.

Line 147: Kind introduce “the” before the “studies”.

Table 1. Tables must always stand alone and be self-explanatory. Consequently, the authors should please indicate what the meaning of the numbers under the “aim” mean to the reader. There are too many abbreviations that are not expanded in Table 1. For example DM, GHGe and DALY. I see that the authors have selected papers that reported on Mediterranean diet for example. The authors could have added search terms including “Mediterranean diet” and “Plant-based diet” during the article search. This is important especially considering that the Title of this systematic review contains “Sustainable Diets”. The “Diets” used in the title is plural.

Synthesis and Discussion

Line 167: “However” should be “however”.

Line 168: Kindly remove one of the “full stop”.

Line 173-174: Could you please provide a reference to support that statement? Why is GHGe the most prominent indicator?

Line 182: I thought the authors were going to compare the GHG footprint with that of plants? If yes, the authors should compare that of the beef, lamb and pork with the  (~1-2 gCO2eq/g) listed for the plants. The same should be done for all the other underlisted footprints.

Line 193: How did they measure the GHGe practically for those group of foods consumed by the participants?

The authors should be clear in the method how the foods contribute towards the GHGe addressed in the discussion. For example, in cows, how do they contribute towards the GHGe? Is because huge tonnes of water are needed to produce a quantity of beef? The amount of methane produced by them as well? How are plant able to reduce the GHGe? Also, the authors could bring to light some advances made in reducing GHGe in cows. For example in New Zealand for instance, there are now efforts to breed low methane-producing cows as a way to reduce their impact on GHGe.

Line 222-223: As this is important, the authors could provide an example of how pollution and environmental deterioration affects our health. This makes the work practical towards its audience

Line 223-236: Kindly provide a reference to support the statement.

Line 234: “partials” for “partial”.

Line 244: How was the assessment done? Through modelling?

Line 267: Do the authors mean “Strategies to promote sustainable diet consumption?” With the strategies underlisted, were they the authors’ subjective proposal or they were systematically selected following a scoping review of the articles selected for the review?  Though the authors indicated in the “Method section” that this work is a “Systematic review” with a “Narrative synthesis” in its approach, more of the strategies could have been identified through a “Thematic analysis” of the papers that were carefully selected.

Conclusion

Line 359: Why would you particularly reference nurses when in the health institutions “Dietitians” and “Nutritionists” should spearhead food-related issues?

Line 361: The use of the calorie-balanced diet based on vegetables a bit vague. The authors should provide clarity on that. What if the diet of the consumer is Medietarranean but instead of vegetables focuses more on legumes, nuts and seeds? Maybe the use of “plant-based diet” is more appropriate? how much do the authors mean by “slight” protein intake? It is good to provide insight into this as present the average protein requirement for a normal healthy person hovers around 0.8g/kg bodyweight. If slight, the authors from the papers they reviewed could provide some form of value so that the reader and the public could practically implement the recommendations made.

References

In all the references cited for the articles from journal publications used for this review, the authors consistently use the date of the publication (day and month) after the journal. That is not correct. The authors should kindly check that and work on it.

However, in the citation of documents from Organizations WHO and FAO for instance, the authors fail to indicate the date the documents were retrieved and cited. Examples include Lines 392-394 and Line 408-409.

Author Response

Dear Reviewer

Round 2

Reviewer 2 Report

Questions have been satisfactorily answered.

Author Response

We greatly appreciate your input
Thanks

Reviewer 3 Report

Thank you for revising the manuscript. There has been a significant improvement in the quality of the content. Please, find below some minor comments for you to address.

Line 23: The authors should please indicate which “Day” in May the review was conducted.

Line 26- 27: Please correct “focusing in fishs and poultry” to “focusing on fish and poultry”. The plural for “fish” is “fishes” and not “fishs”.

Line 60: Kindly correct “precessed” to “processed” if that is the intention of the authors.

Line 120: The authors should kindly indicate the dates i.e the day and month as well.

Line 213-220: The authors should kindly provide a reference to support the statements.

Author Response

Thank you so much for your contributions

greeting
